# Intraperitoneal Administration of 17-DMAG as an Effective Treatment against *Leishmania braziliensis* Infection in BALB/c Mice: A Preclinical Study

**DOI:** 10.3390/pathogens13080630

**Published:** 2024-07-27

**Authors:** Kercia P. Cruz, Antonio L. O. A. Petersen, Marina F. Amorim, Alan G. S. F. Pinho, Luana C. Palma, Diana A. S. Dantas, Mariana R. G. Silveira, Carine S. A. Silva, Ana Luiza J. Cordeiro, Izabella G. Oliveira, Gabriella B. Pita, Bianca C. A. Souza, Gilberto C. Bomfim, Cláudia I. Brodskyn, Deborah B. M. Fraga, Isadora S. Lima, Maria B. R. de_Santana, Helena M. P. Teixeira, Juliana P. B. de_Menezes, Washington L. C. Santos, Patrícia S. T. Veras

**Affiliations:** 1Laboratory of Host-Parasite Interaction and Epidemiology, Gonçalo Moniz Institute, Fiocruz-Bahia, Salvador 40296-710, Bahia, Brazil; kerciapinheirobr@hotmail.com (K.P.C.); petersen.swe@gmail.com (A.L.O.A.P.); marinafaillacea@gmail.com (M.F.A.); alangualbertopinho@gmail.com (A.G.S.F.P.); luanabio@hotmail.com (L.C.P.); diiiana.dantas@gmail.com (D.A.S.D.); marianagueudeville@gmail.com (M.R.G.S.); carinelp@outlook.com (C.S.A.S.); alj.cordeiro@gmail.com (A.L.J.C.); izabella.g1@hotmail.com (I.G.O.); gabriellapita@hotmail.com (G.B.P.); claudia.brodskyn@fiocruz.br (C.I.B.); deborah.fraga@fiocruz.br (D.B.M.F.); mariabrs12@hotmail.com (M.B.R.d.); pitangueirahelena@gmail.com (H.M.P.T.); juliana.fullam@fiocruz.br (J.P.B.d.); 2Baiano Federal Institute of Education, Science and Technology—Santa Inês Campus, BR 420, Santa Inês Road, Rural Zone, Ubaíra 45320-000, Bahia, Brazil; 3Laboratory of Structural and Molecular Pathology, Gonçalo Moniz Institute, Fiocruz-Bahia, Salvador 40296-710, Bahia, Brazil; bianca_arla@hotmail.com (B.C.A.S.); isadoraslima@hotmail.com (I.S.L.); washington.santos@fiocruz.br (W.L.C.S.); 4Laboratory of Population Genetics and Molecular Evolution, Biology Institute, Federal University of Bahia, Salvador 40170-110, Bahia, Brazil; gcbomfim@ufba.br; 5Department of Preventive Veterinary Medicine and Animal Production, School of Veterinary Medicine and Animal Science, Federal University of Bahia, Salvador 40170-110, Bahia, Brazil; 6National Institute of Science and Technology of Tropical Diseases (INCT-DT), National Council for Scientific Research and Development (CNPq); 7Department of Pathology and Forensic Medicine, Bahia Medical School, Federal University of Bahia, Salvador 40110-906, Bahia, Brazil

**Keywords:** *Leishmania braziliensis*, cutaneous leishmaniasis, 17-DMAG-HCl, treatment

## Abstract

Background: Leishmaniasis is a significant global public health issue that is caused by parasites from *Leishmania* genus. With limited treatment options and rising drug resistance, there is a pressing need for new therapeutic approaches. Molecular chaperones, particularly Hsp90, play a crucial role in parasite biology and are emerging as promising targets for drug development. Objective: This study evaluates the efficacy of 17-DMAG in treating BALB/c mice from cutaneous leishmaniasis through in vitro and in vivo approaches. Materials and Methods: We assessed 17-DMAG’s cytotoxic effect on bone marrow-derived macrophages (BMMΦ) and its effects against *L. braziliensis* promastigotes and intracellular amastigotes. Additionally, we tested the compound’s efficacy in BALB/c mice infected with *L. braziliensis* via intraperitoneal administration to evaluate the reduction in lesion size and the decrease in parasite load in the ears and lymph nodes of infected animals. Results: 17-DMAG showed selective toxicity [selective index = 432) towards *Leishmania* amastigotes, causing minimal damage to host cells. The treatment significantly reduced lesion sizes in mice and resulted in parasite clearance from ears and lymph nodes. It also diminished inflammatory responses and reduced the release of pro-inflammatory cytokines (IL-6, IFN-γ, TNF) and the regulatory cytokine IL-10, underscoring its dual leishmanicidal and anti-inflammatory properties. Conclusions: Our findings confirm the potential of 17-DMAG as a viable treatment for cutaneous leishmaniasis and support further research into its mechanisms and potential applications against other infectious diseases.

## 1. Introduction

Leishmaniases, a group of neglected tropical diseases distributed worldwide, are caused by parasites of the *Leishmania* genus that are transmitted to humans through the bite of an insect vector [1]. These diseases can be classified into two main clinical forms: visceral leishmaniasis, characterized by high parasitemia in internal organs, such as the spleen and liver, and tegumentary leishmaniasis, which can present as either a localized or disseminated manifestation [2]. Current leishmaniasis treatments include first-line drugs, such as pentavalent antimonials, and alternative or associated treatments, including amphotericin B, pentamidine, and miltefosine [1,3]. However, these types of chemotherapeutic interventions suffer from limitations, such as an invasive route of administration, prolonged course of therapy, and high cost, all of which contribute to treatment abandonment by patients and therapeutic failure, highlighting the need to develop new drugs to improve treatment options.

Targets, such as the molecular chaperone, which play a crucial role in the functional stability of proteins and cell viability under various environmental stress conditions, such as Hsp90 inhibitors [4,5], can be considered potential treatment options. In protozoan-induced infections, treatment with Hsp90 inhibitors can block parasite proliferation, as the chaperone is essential for differentiation and development at different parasite life cycle stages [5,6]. Hsp90 aids in the cellular response to environmental stimuli, such as changes in pH and temperature, by restoring cellular homeostasis [7,8,9,10].

The Hsp90 has emerged as a potential therapeutic target in diseases caused by diverse parasite genera, including *Eimeria*, *Babesia*, *Theileria*, *Dictyostelium*, *Plasmodium*, *Trypanosoma*, and *Leishmania* [6,9,11]. Geldanamycin, a natural product from the benzoquinone ansamycin family that blocks Hsp90 N-terminal domains, was shown to effectively reduce *T. cruzi* proliferation in a dose-dependent manner [8]. The geldanamycin derivatives, 17-AAG and 17-DMAG, also inhibit this molecular chaperone, and have demonstrated great chemotherapeutic potential against *P. falciparum* strains [12]. Treatment with these compounds causes arrest in the G1 cell cycle phase of epimastigote forms of *T. cruzi*, interfering with conversion towards spheromastigote-like forms [8]. Other Hsp90 inhibitors also exhibited antiparasitic activity against a *T. brucei* species that causes sleeping sickness in humans [13]. Inhibitors that bind to the N-terminal region of Hsp90 were found to be more effective in vitro at nanomolar concentrations, while the novobiocin inhibitor, which binds to the C-terminal region, showed efficacy on a micromolar scale [13].

To further the search for novel therapeutic targets, our group has been studying the chemotherapeutic potential of Hsp90 inhibitors in controlling *Leishmania* infection. Previous studies on the leishmanicidal effect of Hsp90 inhibitors, such as the geldanamycin derivative, 17-AAG, has evidenced a leishmanicidal effect at non-toxic concentrations to human monocytic lineage (THP-1) cells [14]. Notably, 17-AAG effectively controls *L. amazonensis* and *L. braziliensis* infection in vitro [15]. However, this water-insoluble compound partially controls *L. braziliensis* in vivo [16], by reducing parasite load in just 10% in the mouse ear, while no alterations in parasite load in the draining lymph nodes were noted in treated mice compared to controls [16]. To overcome this limitation, the present report endeavored to evaluate the antileishmanial potential of 17-DMAG, a water-soluble analog of 17-AAG, which offers superior pharmacokinetic properties [17,18,19].

Given the urgent need for improved therapeutic approaches for leishmaniasis, our study aims to evaluate the effectiveness of 17-DMAG in a BALB/c mouse model infected with *L. braziliensis*. By utilizing both in vitro and in vivo analyses, we hope to gain valuable insights into its potential as a therapeutic option.

## 2. Materials and Methods

### 2.1. Animals

Female BALB/c mice were maintained in cages at the Animal Care Facility of the Gonçalo Moniz Institute (IGM-FIOCRUZ-BA), with food and water ad libitum. Animals were used in experiments only after reaching an age between six and eight weeks. The present study protocol was approved by the Institutional Review Board on Animal Experimentation of the Gonçalo Moniz Institute (protocols 007/2015 and 007/2020).

### 2.2. Parasites

To maintain infectivity, promastigotes of *L. braziliensis* (MHOM/BR/01/BA788) were periodically isolated from the lymph nodes of infected BALB/c mice. The isolated promastigotes were cultured in Schneider’s complete medium, consisting of Schneider’s insect medium (Sigma-Aldrich, St. Louis, MO, USA) supplemented with 50 μg/mL gentamicin (Sigma-Aldrich) and 20% fetal bovine serum (FBS, Gibco, Waltham, MA, USA). Cultures were incubated in a B.O.D incubator at 24 °C for one week to allow the release of amastigotes from macrophages and differentiation into promastigote forms. These forms were maintained in axenic cultures through successive passages until the seventh passage. For in vitro and some in vivo experiments, upon reaching the stationary phase of growth, axenic promastigotes were submitted to differential centrifugation at 50× *g* for 5 min at 4 °C to separate motile from elongated parasites, the former of which remained in the supernatant following centrifugation. The parasites were then washed three times with saline (0.9% NaCl) by centrifugation at 1560× *g* for 10 min at 4 °C.

In some in vivo experiments, metacyclic forms of *L. braziliensis* enriched by lectin were used to infect mice. To enrich these parasites, cultures of stationary-phase *L. braziliensis* promastigotes were washed with a 0.9% NaCl solution and centrifuged at 1560× *g* for 10 min at 4 °C. The resulting pellet was then resuspended in 1 mg/mL of lectin (Bauhinia Purpurea Lectin—Vector Laboratories, Newark, CA, USA) diluted in 0.9% NaCl solution and incubated for 30 min at room temperature. Following centrifugation at 40× *g* for 5 min at 4 °C, the metacyclic forms were collected from the supernatant. The fraction enriched with the metacyclic forms was washed twice with 0.9% NaCl solution and centrifuged at 2760× *g* for 15 min at 4 °C. After the final wash, the resulting pellet was resuspended in 1 mL of 0.9% NaCl solution, and parasites were homogenized using a 1 mL syringe and insulin needle to disrupt rosettes. Finally, parasites were counted in a Neubauer chamber and then diluted in 0.9% NaCl solution according to the appropriate concentration required for each experimental condition. The yield values of promastigotes after metacyclic enrichment of *L. braziliensis* using the lectin varied from 40.9% to 58.31%, a mean value similar to the yield value previously described by Pinto da Silva et al., 2002 [20].

### 2.3. Isolation and Differentiation of Bone Marrow Precursors into Macrophages (BMMΦ)

To obtain macrophages derived from bone marrow precursors (BMMΦ), BALB/c mice were euthanized with 50 mg/kg of xylazine and ketamine (Syntec, Santana de Parnaíba, SP, Brazil) solution, intraperitoneally, and then mouse femurs and tibias were aseptically removed and placed in ice-cold sterile saline solution containing 0.01 mg/mL of ciprofloxacin (Frenesius Kabi, Bad Homburg, Germany). The bone ends were then removed in a sterile environment, and bone marrow cells were extracted by spraying the bone cavity with complete RPMI medium (Sigma-Aldrich) supplemented with 25 mM HEPES (Sigma-Aldrich), 2 g/L sodium bicarbonate (Sigma-Aldrich), 20% inactivated fetal bovine serum (Gibco), 200 mM glutamine (Sigma-Aldrich), and 0.02 mg/mL ciprofloxacin (Claris, Gujarat, India). The obtained cells were then centrifuged at 250× *g* for 10 min at 4 °C, resuspended, and cultured in disposable bacteriological Petri dishes containing 10 mL of complete RPMI medium supplemented with 30% of L929 fibroblast cell-line culture supernatant enriched with granulocyte–monocyte colony-stimulating factor (GM-CSF), which promotes macrophage differentiation. Cell cultures were incubated at 37 °C under 95% humidity and 5% CO_2_ for 24 h, after which cell supernatants containing macrophage precursors were transferred to new dishes. After an additional 72 h, another 5 mL of complete RPMI medium containing 30% L929 supernatant was added to the cultures. On the seventh day, differentiated macrophages were detached using a 1 mM EDTA (Invitrogen, Carlsbad, CA, USA) solution, and then centrifuged at 250× *g* for 10 min at 4 °C. The obtained cells were then plated as described below.

### 2.4. Assessment of the Cytotoxicity of 17-DMAG against BMMΦ In Vitro

The Alamar Blue^®^ (Invitrogen) reduction assay was used to evaluate BMMΦ viability following exposure to 17-DMAG (Fermentek Ltd., Jerusalem, Israel) (Figure 1) in vitro. BMMΦ were obtained as described above, then distributed on 96-well plates at a concentration of 5 × 10^4^ cells per well containing 100 μL of complete DMEM medium [DMEM low glucose (Gibco) supplemented with 22.8 mM HEPES (Sigma-Aldrich), 27 mM sodium bicarbonate (Sigma-Aldrich), 10% FBS (Gibco), 2 mM glutamine (Sigma-Aldrich), and 10 μg/mL ciprofloxacin]. After 24 h of incubation at 37 °C under 95% humidity and 5% CO_2_, the cell culture medium was replaced with 200 μL of fresh complete DMEM medium containing a range of 17-DMAG concentrations from 50 μM to 0.024 μM at serial 1:2 dilutions, in triplicate. Non-treated cells were used as a control. After 48 h, 10% Alamar Blue^®^ (Invitrogen) (*v*/*v*) was added to the wells and BMMΦ were re-incubated for an additional 24 h. Alamar Blue^®^ reduction was evaluated after 72 h on a SPECTRAmax-340PC spectrophotometer at wavelengths of 570 and 600 nm. The CC_50_ value is identified as the concentration at which 50% of the cells are no longer viable, providing a quantitative measure of cytotoxicity. This value represents the medium of four independent experiments carried out in triplicate.

### 2.5. Assessment of the Antileishmanial Efficacy of 17-DMAG against L. braziliensis Promastigotes In Vitro

To evaluate the effect of 17-DMAG on *L. braziliensis* promastigotes, parasites in the exponential growth phase were plated on 96-well plates at a concentration of 4 × 10^5^ parasites per well containing 100 μL of complete Schneider’s medium. Next, an additional 100 μL of complete Schneider’s medium containing different concentrations of 17-DMAG, ranging from 2 μM to 0.001 μM, were added using a serial 1:2 dilution, in triplicate. After 70 h of incubation in a B.O.D. oven at 24 °C, 10% Alamar Blue^®^ (*v*/*v*) was added to each well, followed by incubation for an additional 2 h. Alamar Blue^®^ reduction was evaluated after 72 h of exposure to the compound using a SPECTRAmax-340PC spectrophotometer at wavelengths of 570 and 600 nm. The percentage of viable cells was plotted against the concentration of 17-DMAG, and the data were fitted to a dose–response curve using a sigmoidal curve fit. The IC_50_ value is identified as the concentration at which 50% of the parasites are no longer viable, providing a quantitative measure of cytotoxicity. This value represents the medium of four independent experiments carried out in triplicate.

### 2.6. Evaluation of the Antileishmanial Efficacy of 17-DMAG against Intracellular Amastigote of L. braziliensis In Vitro

BMMΦ from BALB/c mice were distributed across 24-well plates at a concentration of 2 × 10^5^ cells in 1 mL of complete DMEM medium, then incubated for four hours at 37 °C under 5% CO_2_ and 95% humidity. After three washes with saline solution to remove non-adherent cells, *L. braziliensis* promastigotes in the stationary phase of growth were obtained as described in Section 2.2 and then added to BMMΦ cultures at a ratio of 10:1. After 24 h, non-internalized parasites were subsequently removed by washing. Infected cells were then treated with different concentrations of 17-DMAG, ranging from 900 to 1 nM, following 1:2 serial dilution steps, in quintuplicate. After 72 h of treatment, the wells were washed twice with saline solution and 1 mL of complete Schneider’s medium was replaced in each well. The plates were then incubated for 6 days in a B.O.D. oven at 24 °C to allow viable amastigotes to differentiate into promastigote forms. Finally, viable promastigotes were randomly counted in a Neubauer chamber. The percentage of remaining infected cells was plotted against the 17-DMAG concentration, and the data were fitted to a dose–response curve, using a sigmoidal curve. The IC_50_ value was identified as the concentration at which the activity is reduced by 50%, providing a quantitative measure of inhibitory potency. This value represents the medium of four independent experiments carried out in triplicate.

The selectivity index of 17-DMAG was calculated as the ratio between the CC_50_ for bone marrow-derived macrophages (BMMØ) and the IC_50_ for *L. braziliensis* promastigotes and intracellular amastigotes (CC_50_/IC_50_), providing a measure of the drug’s efficacy against the promastigote and intracellular amastigote forms compared to its toxicity to host cells.

### 2.7. Long-Term Post-Treatment Observation of Intraperitoneally Administered 17-DMAG to L. braziliensis Infected BALB/c Mice

BALB/c mice were infected in the ear with 10^5^ promastigotes of *L. braziliensis* in the stationary phase. The parasites were obtained as described in Section 2.2, suspended in 10 µL of saline solution and inoculated in the ear of BALB/c mice aged 6 to 8 weeks using a microsyringe (BD Ultra-Fine™ II—0.5 mL). After two weeks of infection, when a papule appeared at the inoculation site, the ear thickness of the animals was measured using a manual caliper (Köeppen—accuracy ± 0.1 mm), and then they were equally distributed in the different experimental groups, based on their lesion sizes. To determine the optimal dose/regimen to be used intraperitoneally in BALB/c mice, groups of six mice were submitted to 30-day 17-DMAG treatment using three different regimens: 20 mg/kg daily, 30 mg/kg every 2 days, and 50 mg/kg every 5 days. Once the optimal dose/regimen was determined, additional experiments were performed using 20 mg/kg daily for different time points, including long-term post-treatment evaluation at 2, 4, or 7 weeks. In these experiments, the control group, containing at least five animals, received an intraperitoneal injection of a 5% glucose solution (Isofarma Industrial Farmacêutica LTDA, Eusébio, CE, Brazil) in a volume equivalent to that used for the 17-DMAG solution. The ear thickness of the mice was measured weekly using a manual caliper, and results were expressed as the difference between lesion thickness and the thickness of the contralateral ear. The area under the curve (AUC) was calculated to analyze lesion progression in the ears of treated and untreated *L. braziliensis*-infected mice over time using the trapezoidal method. Lesion clinical appearances were evaluated throughout the treatment. After each assay, the animals were euthanized with 50 mg/kg of xylazine and ketamine solution, intraperitoneally, and their ears were removed for histopathological analysis, or the ears and lymph nodes were aseptically removed for determining the parasitic load.

### 2.8. Evaluation of the Effect of 17-DMAG Treatment on the Inflammatory Infiltrate in Lesions of L. braziliensis-Infected BALB/c Mice

BALB/c mice were infected with 5 × 10^5^ promastigotes of *L. braziliensis* in stationary phase enriched with metacyclic forms, obtained as described in Section 2.2. After three weeks of infection, the animals were equally distributed into two groups based on ear thickness; one group was treated with a daily dose of 20 mg/kg of 17-DMAG intraperitoneally, and the control group was treated with a 5% glucose solution, as described in Section 2.7. At 2, 4, 7, 14, or 28 days of treatment, the animals were euthanized, and their ears were extracted for histopathological analysis.

### 2.9. Evaluation of 17-DMAG Treatment on Pro-Inflammatory Cytokine Release by Lymph Node Cells from L. braziliensis-Infected BALB/c Mice

After two and four weeks after daily treatment with 20 mg/kg of 17-DMAG, mice were euthanized, as explained in Section 2.7, and retroauricular lymph nodes from these animals were aseptically removed and homogenized. Lymph nodes cells were then plated in complete RPMI medium at a concentration of 10^6^ cells per well. These cells were restimulated with *L. braziliensis* (in a proportion of five parasites for each lymphocyte) or 5 µg/mL of concanavalin A (ConA, Sigma-Aldrich) for 48 h. After this period, the supernatants were collected, centrifuged at 375× *g* for 10 min at 4 °C, and kept at −20 °C until use. The production of the cytokines IL-6, IL-10, IL-12, TNF, and IFN-γ and of the chemokine MCP-1 in cells’ supernatant was evaluated using a specific CBA kit for inflammatory cytokines (BD Biosciences, San Jose, CA, USA) according to the manufacturer’s instructions.

### 2.10. Quantification of Parasite Burden in BALB/c Mice Infected with L. braziliensis Using Limiting Dilution Assay

After two, four, or seven weeks of daily treatment with 20 mg/kg of 17-DMAG, mice in control or treated groups were euthanized, as described in Section 2.7, and their ears and lymph nodes were removed aseptically. The tissues were then macerated to obtain infected cells, which were subsequently centrifuged at 300× *g* for 10 min at 4 °C, and the resulting pellet was resuspended in complete RPMI medium. Next, 20 μL of each tissue suspension was dispensed in triplicate into a 96-well plate previously containing 180 μL of complete Schneider’s medium. Finally, an eight-step serial dilution was performed. Plates were then incubated in a B.O.D. incubator to assess parasite growth by optical microscopic observations every 3 days for 15 days. At the end of the observation time, a value of 10^1^ was assigned for the first three positive wells. For the subsequent positive wells of the dilution, the power increased 10 times, exponentially.

### 2.11. Histopathological Characterization of Lesions in BALB/c Mice Infected with L. braziliensis and Treated with 17-DMAG

Tissues extracted from BALB/c mice infected or infected and treated daily with 17-DMAG, as described in Section 2.7 and Section 2.8, were incubated in acid formalin solution (5% glacial acetic acid (Sigma-Aldrich)), 10% formaldehyde (Sigma-Aldrich), and 85% absolute ethanol (Sigma-Aldrich) for 48 h to maintain tissue integrity. The samples were then transferred to the Histotechnology Platform of the Gonçalo Moniz Institute for processing, where they were maintained in 70% alcohol until paraffin embedding and cut into thin, uniform sections. These sections were then stained with hematoxylin and eosin (HE stain) according to the Platform’s protocol and the tissues were analyzed using light microscopy.

### 2.12. Statistical Analysis

When data presented a Gaussian distribution, Student’s *t*-test was used to compare two groups, and one-way ANOVA followed by Tukey’s post hoc test was used to compare three or more groups. For non-Gaussian distribution, the Mann–Whitney U test was used to compare two groups, and the Kruskal–Wallis test was used to compare three or more groups. Differences were considered statistically significant when the *p*-value was <0.05. GraphPad Prism version 8.0 software (GraphPad Software Inc., La Jolla, CA, USA) was used for statistical analysis and graph generation.

## 3. Results

### 3.1. Assessment of the 17-DMAG Effectiveness against L. braziliensis In Vitro

First, the evaluation of 17-DMAG toxicity on BMMΦ was conducted, and it revealed that only at the highest concentrations of 25 and 50 μM did the drug cause 100% cell death, while at 12.5 μM, it was lethal for more than 96% of exposed cells (Figure 2A). From 0.024 μM to 6.25 μM, a gradual decrease in cell viability was observed. After 72 h of exposure, the concentration capable of causing toxicity to 50% of macrophages (CC_50_) was found to be 3.2 ± 0.47 μM (Figure 2A).

The efficacy of 17-DMAG was first tested against axenic promastigotes of *L. braziliensis*, which caused, respectively, the death of 88.66% and 93.12% of the parasites at concentrations of 1 and 2 μM (Figure 2B). At concentrations ranging from 0.00097 to 0.5 μM, all parasite viability values were nearly 100% (Figure 2B). The concentration of 17-DMAG capable of causing the death of 50% of the parasites in their promastigote form (IC_50_) was found to be 0.67 ± 0.04 μM. These results show that 17-DMAG is almost five times more toxic to *L. braziliensis* promastigotes than to macrophages.

The in vitro tests to evaluate the effectiveness of 17-DMAG against intracellular parasites residing within macrophages of mammalian hosts were subsequently conducted. Our findings revealed that treating *L. braziliensis*-infected BMMΦ with 17-DMAG resulted in an IC_50_ value of 7.4 ± 1.64 nM, 90 times lower than the concentration required to kill the free form of promastigotes that inhabit the sand fly vector. As illustrated in Figure 2C, BMMΦ treatment at concentrations ranging from 15.6 to 1000 nM led to the complete clearance of intracellular parasites. Only at concentrations between 0.9 to 7.8 nM did we observe a gradual reduction in parasite viability, revealing more than 400 times lower toxicity in comparison to BMMΦ, resulting in a selective index of 432.43.

### 3.2. Evaluation of Intraperitoneally Administered 17-DMAG Treatment against L. braziliensis Infection in BALB/c Mice

The most effective intraperitoneal (IP) treatment dose for 17-DMAG was determined by testing different regimens of 17-DMAG on *L. braziliensis*-infected BALB/c mice. At the end of thirty days of treatment, compared to untreated mice (0.6653 ± 0.3371 mm), a 50 mg/kg dose every five days resulted in no reduction in lesion size (0.5225 ± 0.2572, one-way ANOVA, *p* = 0.6994) (Figure 3A). However, mice treated with 30 mg/kg of 17-DMAG every other day showed a significant reduction in lesion size in comparison to the control group (0.4447 ± 0.1744, one-way ANOVA, *p* = 0.0081) (Figure 3A). The most impressive effect was seen with the dose of 20 mg/kg daily, which induced a higher and more significant reduction in lesion size when compared to the control group in the 30th day of treatment (0.3696 ± 0.1262, one-way ANOVA, *p* < 0.0001) (Figure 3A). Notably, the decrease in ear thickness of these mice was more significant after the second week of treatment. To analyze the lesion progression in the ears of animals treated with different regimens over time, we used the AUC method. Considering the 4 weeks of follow-up, all treatment regimens resulted in AUCs that were smaller in comparison to the AUC of untreated mice (*p* = 0.0001, one-way ANOVA, Figure 3B). However, the AUCs of animals treated with the 20 mg/kg (*p* < 0.0001) or 30 mg/kg (*p* = 0.0016) regimens showed more pronounced differences from those of the control mice (Figure 3B) and from that produced by the 50 mg/kg every-five-days regimen compared to the AUC of controls (*p* = 0.0373) (Figure 3B).

Figure 4 depicts lesion aspects of the ears from infected mice, clearly showing a difference between treated animals and the control group at all evaluated time points. Ears from treated mice show smaller or no lesions (Figure 4E–P), while untreated mice present well-developed lesions (Figure 4A–D). Animals treated with 20 mg/kg daily (Figure 4E–H) or 30 mg/kg every 2 days (Figure 4I–L) exhibited few or no lesions. In contrast, those treated with 50 mg/kg every five days (Figure 4M–P) showed less inflamed lesions than untreated mice (Figure 4A–D), yet these lesions were more developed than those in mice treated with the other two regimens and were accompanied by intense redness (Figure 4E–P). These findings indicate that intraperitoneal administration of 17-DMAG, using regimens of either 20 mg/kg daily or 30 mg/kg every two days, not only reduces lesion size caused by *L. braziliensis* but also diminishes the local inflammatory response. These results suggest that more frequent administration at shorter intervals proved to be necessary to optimize the effectiveness of 17-DMAG treatment.

Since the most effective results in mice were obtained with a daily dose of 20 mg/kg, we selected this regimen for the subsequent experiments. We conducted a kinetic study and observed an intense reduction in lesion size starting from the second week of treatment. After two weeks of treatment, the median ear thickness in the treated group was 3.7 times smaller than that of the control group (*p* = 0.0065). By the fourth week, this difference was 7.2 times smaller for the treated group compared to the control group (*p* = 0.0022). At the fifth week, the difference increased to 14.8 times smaller for the treated group compared to the control animals (*p* = 0.0022). During the fifth week, peak lesion size was observed in the control group, while from the sixth to the eighth week, lesion sizes from treated animals showed no differences compared to contralateral uninfected ears (Mann–Whitney, Figure 5A). The lesion progression analysis showed that the median AUC of lesion thickness in the treated group was 4.2 times smaller than that in the control group [0.65 (Q1: 0.54; Q3: 0.83) vs. 2.74 (Q1: 1.41; Q3: 3.56), Mann–Whitney, *p* < 0.0001] (Figure 5B).

The parasite burden in the ears and retroauricular lymph nodes of these BALB/c mice infected with *L. braziliensis* and treated with 17-DMAG was markedly lower than in the control group at all assessed time points. After two weeks of treatment, the infected and treated mice exhibited the presence of a five-log difference in parasite numbers in the ears (Figure 6A) and lymph nodes (Figure 6B) of treated compared to untreated animals. After four weeks of treatment, 17-DMAG successfully eliminated the parasite load in the ears of 6 out of 7 animals, while the control group exhibited a high parasite load with a median of 1 × 10^6^ parasites (Q1: 1 × 10^4^; Q3: 1 × 10^8^) (*p* < 0.01) (Figure 6C). Similarly, in the lymph nodes, 17-DMAG eliminated the parasite load in 5 out of 7 animals treated for four weeks, whereas the control group maintained a load of approximately 1 × 10^5^ parasites (Q1: 1 × 10^5^; Q3: 1 × 10^8^) (*p* < 0.001) (Figure 6D).

After seven weeks of treatment, no parasites were detected in the ears (Figure 6E) and lymph nodes (Figure 6F) of the treated animals. Given that the BALB/c mouse model is known to control parasite loads at the infection site [21], there was no statistical difference observed in the parasite load in the ears of the treated animals compared to the control group (Figure 6E). Notably, in the lymph nodes, it was observed that treatment with 17-DMAG successfully eliminated the parasite load in all the treated animals, while in the control group, the parasite load persisted at around 5.5 × 10^4^ parasites (*p* < 0.01) (Figure 6F).

Representative macroscopic images of lymph node and lesion sizes in the ears of animals treated or not treated with 17-DMAG over seven weeks are shown in Figure 7. After two weeks of intraperitoneal treatment with 20 mg/kg/day of 17-DMAG, the ear lesion size (Figure 7G) and lymph node size (Figure 7H) were already smaller than in the control group (Figure 7A and Figure 7B, respectively). This trend was observed also after four (Figure 7I and Figure 7J vs. Figure 7C and Figure 7D, respectively) and seven weeks of treatment (Figure 7K and Figure 7L vs. Figure 7E and Figure 7F, respectively). The treated group exhibited almost imperceptible ear lesions (Figure 7G,I,K) and a progressive decrease in lymph node size (Figure 7H,J,L) throughout the treatment compared to the control group (lesion size, Figure 7A,C,E, lymph node size, Figure 7B,D,F).

Regarding the animals’ clinical appearance, signs of toxicity, such as dehydration, closed eyes with secretion, and dry stools, were observed only in animals treated with 17-DMAG for a longer period (seven weeks). These results demonstrate that 17-DMAG can be effective for the treatment of experimental cutaneous leishmaniasis but exhibits some degree of toxicity when used for prolonged periods.

Histopathological analyses of lesion and lymph node tissues show that in treated animals we observed a complete resolution of the inflammatory process as early as 2 weeks after treatment initiation (Figure 8D). By contrast, in control groups, an inflammatory process ranging from moderate (Figure 8A,B) to intense (Figure 8C) was observed during the evaluated time points. After two weeks, control mice exhibited an inflammatory infiltrate in the ear mainly composed of neutrophils and macrophages (Figure 8A). After four (Figure 8B) and seven weeks (Figure 8C), the ear lesion of the control group was characterized by a predominance of lymphocyte and mononuclear cells, and at the end of the seventh week of treatment, these animals also presented epithelioid cells in the inflammatory infiltrate of infected ears.

Taken together these findings suggest that treatment with 17-DMAG together with its high antileishmanial effect leads to an intense reduction in the inflammatory process at the site of the *Leishmania* lesion as early as two weeks of treatment. These characteristics highlight the drug’s anti-inflammatory potential, as well as an in vivo antileishmanial agent when administered intraperitoneally.

To monitor the temporal efficacy of 17-DMAG in controlling the inflammatory process, we assessed histopathological alterations at time points preceding the two-week treatment interval, specifically at 2, 4, and 7 days. Additionally, evaluations were conducted at the previously analyzed time points of 14 and 28 days. After 2 (Figure 9A,K) or 4 days of treatment (Figure 9C,M), we observed no significant difference between the ear thicknesses of treated and untreated animals, as depicted in Figure 9U for 2 days of treatment (0.34 ± 0.14 vs. 0.25 ± 0.07, *p* = 0.37) and in Figure 9V for a 4 day-regimen (0.41 ± 0.22 vs. 0.37 ± 0.25 *p* = 0.83). However, after 7 (Figure 9E,O), 14 (Figure 9G,Q), or 28 days of treatment (Figure 9I,S), the ear thickness of animals in the treated group was significantly smaller than that of animals in the control group, as shown in Figure 9W (7 days: 0.25 ± 0.20 vs. 0.67 ± 0.41, *p* = 0.015), Figure 9X (14 days: 0.16 ± 0.16 vs. 0.73 ± 0.40, *p* = 0.009), and Figure 9Y (28 days: 0.05 ± 0.05 vs. 0.76 ± 0.18, *p* = 0.003). Evaluation of the ear thickness progression at the 28th day of treatment revealed that the calculated AUC of the treated group (0.60 ± 0.60) was 5.5 times smaller than that of the animals of control group (3.30 ± 0.58, *p* = 0.005). Additionally, lymph nodes from infected and treated animals were reduced in size compared to those from control groups at 7 days (Figure 9F,P), 14 days (Figure 9H,R), and 28 days (Figure 9J) of treatment.

The impact of 17-DMAG treatment on the inflammatory response in mice infected with *L. braziliensis* was investigated through histopathological examination of their ears at earlier time points after treatment. After two days of treatment, a reduction in inflammatory infiltrates was observed, even before any detectable changes in lesion size (Figure 10F). At this initial stage, the predominant cell types were polymorphonuclear cells and lymphocytes. By day seven, a significant decrease in the inflammatory process was noted in the ears of treated mice (Figure 10H) compared to the control group (Figure 10C), marked by a reduced presence of macrophages. By day 28, the treated group showed almost no inflammation (Figure 10J), while the control group still exhibited intense mononuclear inflammatory infiltrates. These results highlight that the antileishmanial effects of 17-DMAG are closely linked to its anti-inflammatory properties when administered intraperitoneally in *L. braziliensis*-induced lesions.

### 3.3. Evaluation of 17-DMAG Treatment on Pro-Inflammatory Cytokines Released by Lymph Node Cells from L. braziliensis-Infected BALB/c Mice

Lymph node cells obtained from animals treated with 17-DMAG for a two-week duration (Figure 11A) and subsequently restimulated in vitro with *L. braziliensis* exhibited significantly reduced production of various proinflammatory cytokines. IL-6 production by lymphocytes derived from 17-DMAG-treated animals was 5.3 times lower (23.36 ± 13.61 pg/mL) compared to lymph node cells from control animals (125.10 ± 62.73 pg/mL) (*p* = 0.0076). IFN-γ release by lymphocytes from animals treated for two weeks was 16.1 times lower (219.10 ± 136.20 pg/mL) compared to the production by lymphocytes from control animals (3536.00 ± 672.20 pg/mL) (*p* < 0.0001). Similarly, TNF production by lymphocytes from treated mice was 4.5 times lower (125.40 ± 48.70 pg/mL) than that observed in lymphocytes from control animals (568.50 ± 94.48 pg/mL) (*p* < 0.0001). The modulatory cytokine IL-10 was also released in lower amounts (475.10 ± 372.30 pg/mL) compared to in untreated animals (2070.00 ± 334.60 pg/mL) (*p* < 0.0001). No significant difference was observed in MCP-1 production between the treated (74.95 ± 2.94 pg/mL) and control groups (74.36 ± 2.55 pg/mL) after two weeks of treatment (*p* = 0.74).

Cells from animals treated for a duration of 4 weeks (Figure 11B) also exhibited significantly reduced production of the pro-inflammatory cytokines previously tested at the 2-week time point (Figure 11A). Lymphocytes from treated mice also demonstrated no difference of MCP-1 between control and treated groups. These results clearly highlight the predominant inhibitory effect of 17-DMAG on the production of inflammatory cytokines.

## 4. Discussion

In this study, we evaluated the efficacy of 17-DMAG, an Hsp90 inhibitor, in controlling *L. braziliensis* infections both in vitro and in vivo through intraperitoneal administration. Initial tests revealed that 17-DMAG is more toxic to *L. braziliensis* than to murine bone marrow-derived macrophages, with selectivity index (SI) values of 4.75 for promastigote forms and 432.4 for intracellular amastigotes. We demonstrated that this drug is highly effective at inhibiting the growth of axenic *L. braziliensis*, even at nanomolar concentrations significantly lower than those that are toxic to host cells.

Supporting the present study, Palma et al. (2019) previously reported that Hsp90 inhibitors, including geldanamycin and its synthetic derivatives 17-AAG and 17-DMAG, reduced the number of *L. amazonensis* promastigotes by 50% at concentrations much lower than the CC_50_ for THP-1 macrophage lines [14]. Petersen et al. (2012) also demonstrated that 17-AAG, a water-insoluble analog of 17-DMAG-HCl, was effective against *L. amazonensis* in CBA macrophages without showing host cell toxicity, achieving a selectivity index (SI) close to 100 [15]. Furthermore, Santos et al. (2014) observed effective in vitro control of *L. braziliensis* in BALB/c macrophages using 17-AAG, reaching a SI around 20 [16]. This index is twenty times lower than the SI observed for 17-DMAG-HCl against *L. braziliensis* intracellular amastigotes described in the present report. Our current results also emphasize that Hsp90 inhibitors have a stronger affinity for *Leishmania* Hsp90 compared to their affinity for host cells.

In this study, we observed a significant reduction in parasitic load compared to controls, along with low toxicity to host cells in vitro. Previous research has shown that Hsp90 inhibitors from the geldanamycin family are effective against intestinal and protozoan parasites [12,14,15,22,23,24]. Both the current and prior in vitro studies suggest that these inhibitors have a higher affinity for the parasite Hsp90 protein compared to the host Hsp90, which may explain the increased susceptibility of parasites due to the binding of inhibitors to the N-terminal domain of leishmanial-Hsp90 [12,14]. The mechanism by which protozoan parasites are killed involves overloading the ubiquitin–proteasome system, which leads to an accumulation of ubiquitin-labeled proteins that ultimately results in incidental parasite death [25]. The importance of targeting Hsp90 in parasites has also been demonstrated with other protein inhibitors against *Plasmodium falciparum*. These inhibitors kill parasites by disrupting Hsp90 stability through their interaction with the active proteasome 26S complex [26].

In vivo assessments of 17-DMAG-HCl showed that it has low toxicity and high efficacy, as demonstrated by significant early reductions in ear thickness, lymph node size, and tissue inflammatory infiltrate in treated BALB/c mice compared to controls. We also observed a clearance in parasitic load compared to controls following treatment. These findings highlight not only the leishmanicidal properties of chlorhydrate of 17-DMAG but also its strong anti-inflammatory effects, which help mitigate the characteristic lesions associated with experimental cutaneous leishmaniasis. The anti-inflammatory properties of geldanamycin and its analogs, including 17-DMAG, have been established in various models of inflammatory diseases [27,28]. In particular, the anti-inflammatory effects of 17-DMAG are mediated through multiple mechanisms, as demonstrated by its impact on human pro-inflammatory T cell responses [29]. Non-toxic concentrations of 17-DMAG effectively suppress T cell proliferation and cytokine secretion, suggesting that Hsp90 inhibitors may be useful in treating diseases characterized by unregulated Th1 and Th17 responses [30]. Treatment with 17-DMAG also leads to decreased expression of Akt and IKK, reduced nuclear translocation of NF-κB, and lowered production of key pro-inflammatory cytokines, such as IL-6, TNF, and NO, thus inhibiting the NF-κB-mediated inflammatory cascade [31,32]. Further studies have shown an early reduction in inflammatory recruitment at the site of infection, likely due to the drug’s effects on pro-inflammatory cytokines and chemokines, like IL-6, TNF, and MCP-1 [15]. Additionally, 17-AAG induces apoptosis by promoting the mitochondrial release of cytochrome C and Smac/DIABLO, activating caspase-9 and caspase-3, and causing a conformational change in Bax, underscoring its potential therapeutic uses in cancer treatment [33].

## 5. Conclusions

Our study underscores the substantial promise of 17-DMAG-HCl as a therapeutic option for treating cutaneous leishmaniasis, given its potential to clear parasitic load and modulate the inflammatory host response. However, further research is necessary to explore the mechanisms involving the modulation of cytokine profiles induced by 17-DMAG-HCl treatment in vivo. These studies will enhance our understanding of its therapeutic efficacy and to pave the way for designing new treatment strategies incorporating this compound for cutaneous leishmaniasis. Additionally, verifying the potential applications of 17-DMAG-HCl in treating various forms of leishmaniasis is an interesting perspective.

## Figures and Tables

**Figure 1 pathogens-13-00630-f001:**
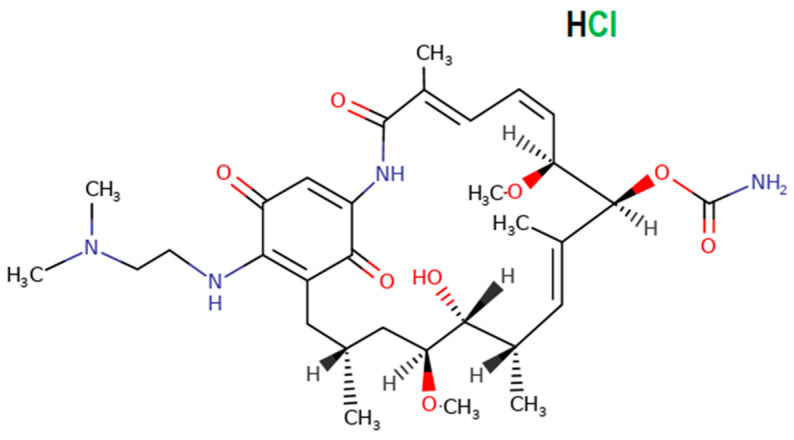
Chemical structure of 17-DMAG-hydrochloride (Fermentek Ltd.). Available at https://www.fermentek.com/product/17-dmag-hydrochloride (accessed on 15 August 2023).

**Figure 2 pathogens-13-00630-f002:**
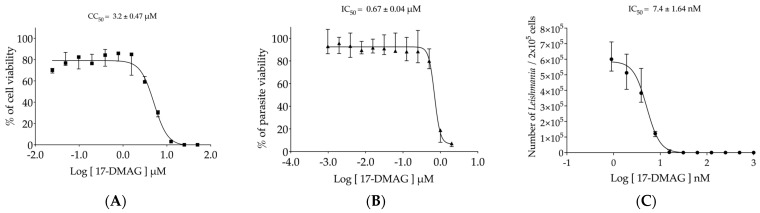
Evaluation of 17-DMAG’s effectiveness against *Leishmania* in vitro. (**A**) Cytotoxicity of 17-DMAG against BMMΦ in vitro. BMMΦ were plated at a concentration of 5 × 10^4^ cells per well in a 96-well plate and incubated with different concentrations of 17-DMAG (0.024 to 50 μM) in 200 µL of DMEM low complete medium. The graph represents the median and quartiles values of the % of viable macrophages from one representative of five independent experiments carried out in triplicate, and CC_50_ was calculated as the mean ± SE of five experiments. (**B**) Efficacy of 17-DMAG against axenic promastigotes of *L. braziliensis*. Axenic promastigotes of *L. braziliensis* in the exponential growth phase were plated at a concentration of 4 × 10^5^ parasites per well in a 96-well plate and incubated with various concentrations ranging from 0.001 μM to 2 μM of 17-DMAG in 200 µL of complete Schneider’s medium. The graph shows the median of the % of viable parasites treated with various 17-DMAG concentrations from one out of four independent experiments carried out in triplicate, and the IC_50_ value is given by the mean ± SE of four experiments. (**C**) Effectiveness of 17-DMAG in controlling BMMΦ infection by *L. braziliensis*. BMMΦ were infected with *L. braziliensis* at a 10:1 ratio. After 24 h, the infected cells were treated with different concentrations of 17-DMAG (0.9 to 900 nM) or left untreated. Viable parasites were assessed after 6 days of differentiation. The graph represents the median and interquartile values of the number of intracellular viable parasites liberated from treated macrophages from one out of four independent experiments conducted in quintuplicate. The IC_50_ value was calculated as the mean ± SE of four experiments. For details, see Section 2.4, Section 2.5 and Section 2.6.

**Figure 3 pathogens-13-00630-f003:**
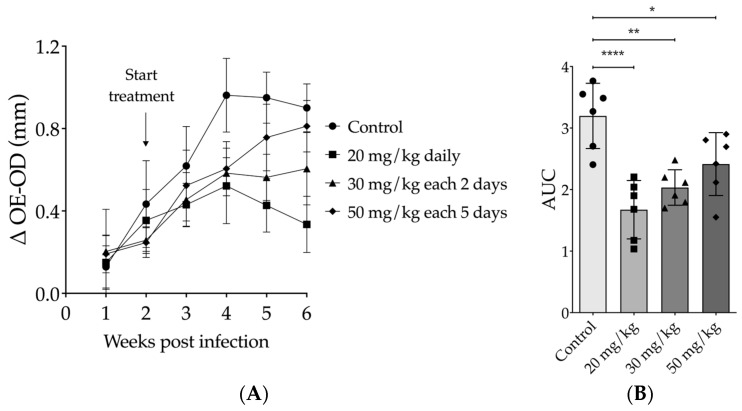
Effect of intraperitoneal treatment with 17-DMAG in different therapeutic regimens on the development of lesions in mice infected with *L. braziliensis*. BALB/c mice were infected in the ear with 10^5^ promastigotes of *L. braziliensis* in the stationary phase. After two weeks of infection, mice were treated intraperitoneally with 17-DMAG at doses of 20 mg/kg daily (n = 6), 30 mg/kg every two days (n = 6), or 50 mg/kg every five days (n = 6). Untreated control animals received 5% glucose solution every day intraperitoneally. (**A**) Graph depicts the difference of thickness in mm (Δ OE-OD) measured from the infected ear compared to the contralateral ear over four weeks of treatment; (**B**) area under the curve (AUC) of the thickness of the lesions shown in (**A**). One-way ANOVA test, * *p* < 0.05; ** *p* < 0.005; **** *p* < 0.0001.

**Figure 4 pathogens-13-00630-f004:**
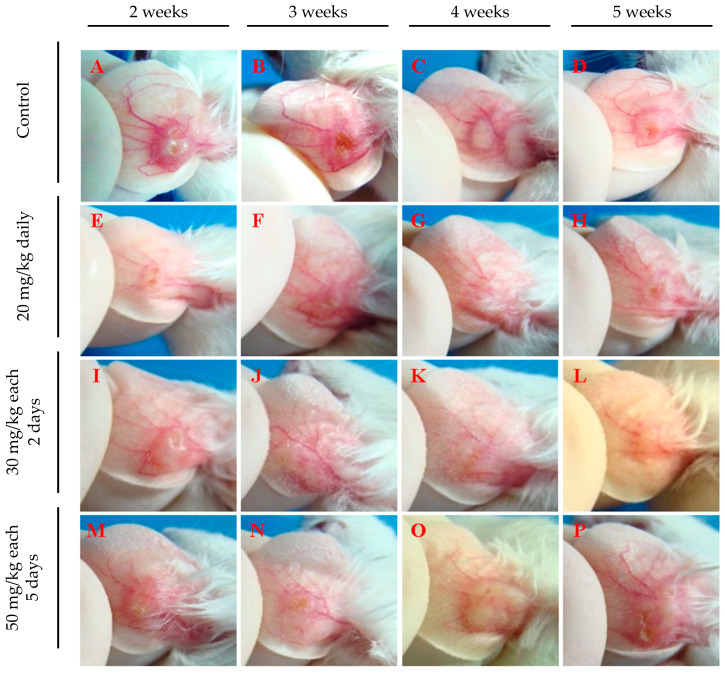
Representative images of ear lesions of *L. braziliensis*-infected BALB/c mice intraperitoneally treated with different therapeutic regimens of 17-DMAG. BALB/c mice were infected in the ear with 10^5^ stationary phase promastigotes of *L. braziliensis* and, after two weeks, animals were intraperitoneally treated for 30 days, with the following dose regimens: (**E**–**H**) 20 mg/kg daily, (**I**–**L**) 30 mg/kg every two days, or (**M**–**P**) 50 mg/kg every five days. (**A**–**D**) Control mice received a daily intraperitoneally injection of 5% of glucose solution. Images were obtained from each animal in the experimental groups on the 30th day of treatment.

**Figure 5 pathogens-13-00630-f005:**
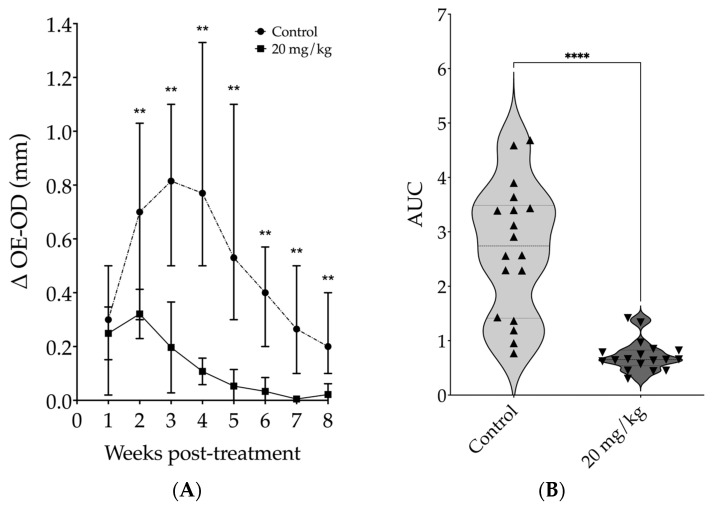
Assessment of kinetics from daily intraperitoneal treatment with 20 mg/kg of 17-DMAG on lesion development in mice infected with *L. braziliensis*. BALB/c mice were infected in the ear with 10^5^ promastigotes of *L. braziliensis* in the stationary phase. After two weeks of infection animals were treated with 20 mg/kg/day of 17-DMAG for seven weeks, intraperitoneally, or with 5% of glucose solution, as control. The results are expressed as the median of the difference between the thickness of the infected ear and that of the contralateral ear. (**A**) Weekly thickness of the lesion in the ear of the animals treated or not with 17-DMAG, Mann-Whitney test, ** *p* < 0.05; (**B**) area under the curve (AUC) of the thickness of the lesions shown in (**A**), Mann–Whitney test, **** *p* < 0.0001. Graphs represent the median and quartiles of one experiment representative of 3 independent experiments conducted in at least sextuplicate and each point represents the AUC of each animal from control and treated groups.

**Figure 6 pathogens-13-00630-f006:**
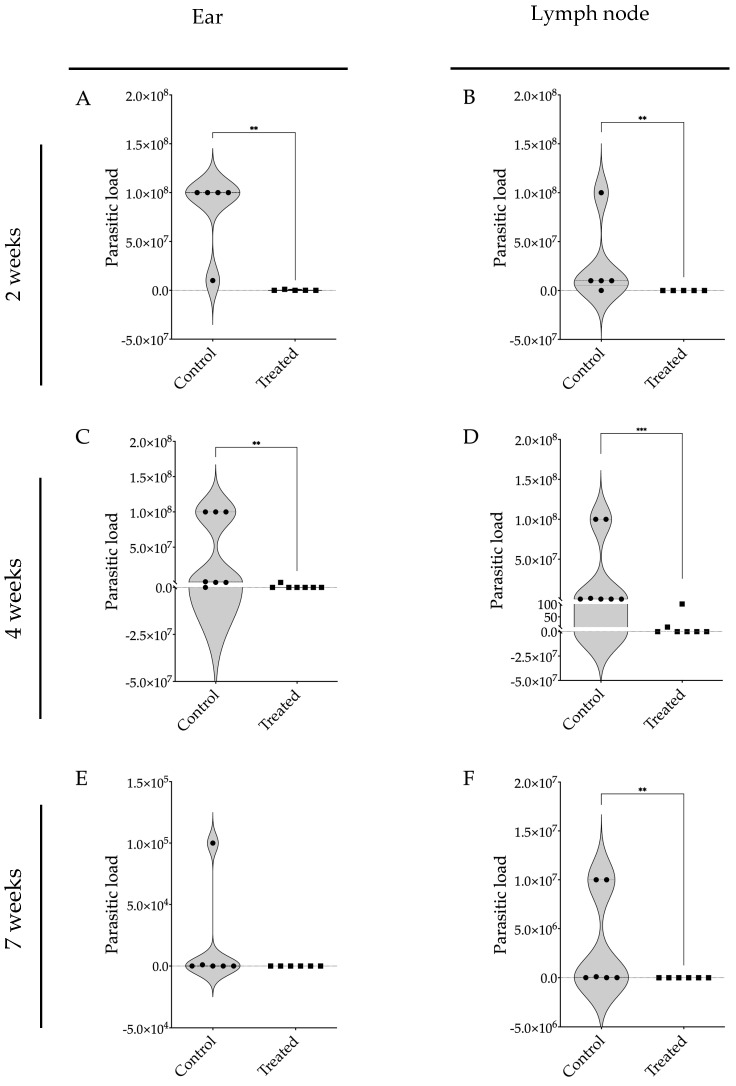
Parasite burden in BALB/c mice infected with *L. braziliensis* and treated with 17-DMAG. BALB/c mice were infected in the ear with 10^5^ promastigotes in the stationary phase of *L. braziliensis* and, after two weeks, treated intraperitoneally with 20 mg/kg/day of 17-DMAG or with 5% of glucose solution, as control, for two (**A**,**B**), four (**C**,**D**), or seven weeks (**E**,**F**). After each time point, the animals were euthanized and the ears (**A**,**C**,**E**) and retroauricular lymph nodes (**B**,**D**,**F**) were removed, macerated and the infected cells were plated into a 96-well plate to evaluate the parasitic burden in these organs using a limiting dilution assay. Mann–Whitney test, ** *p* < 0.01, *** *p* < 0.001.

**Figure 7 pathogens-13-00630-f007:**
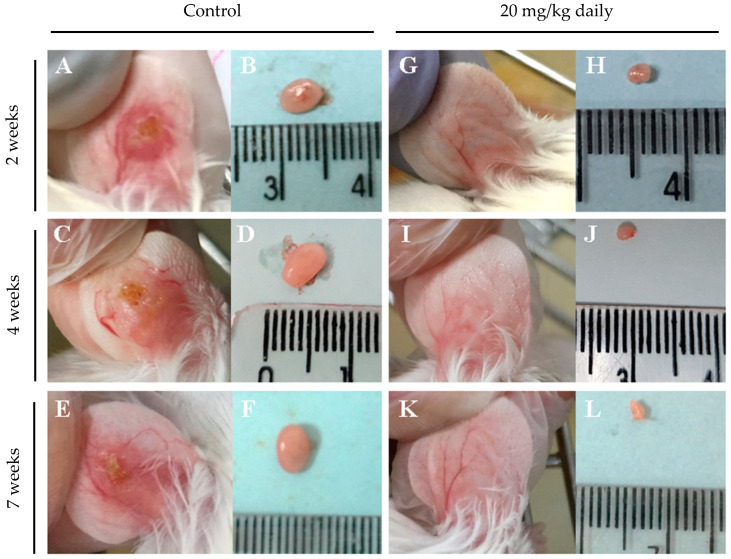
Macroscopic aspect of lesions in the ear and of lymph nodes from BALB/c mice infected with *L. braziliensis* and intraperitoneally treated with 17-DMAG for seven weeks. Images represent lesion and lymph nodes of BALB/c mice infected (**A**–**F**) or infected and treated (**G**–**L**) in the kinetic experiment described in Figure 5. After two weeks of infection, animals were intraperitoneally treated with 20 mg/kg daily of 17-DMAG for two (**G**,**H**), four (**I**,**J**), or seven (**K**,**L**) weeks and euthanized for macroscopic analysis. As controls, mice received 5% glucose solution intraperitoneally during the same time points: two (**A**,**B**), four (**C**,**D**), or seven (**E**,**F**) weeks.

**Figure 8 pathogens-13-00630-f008:**
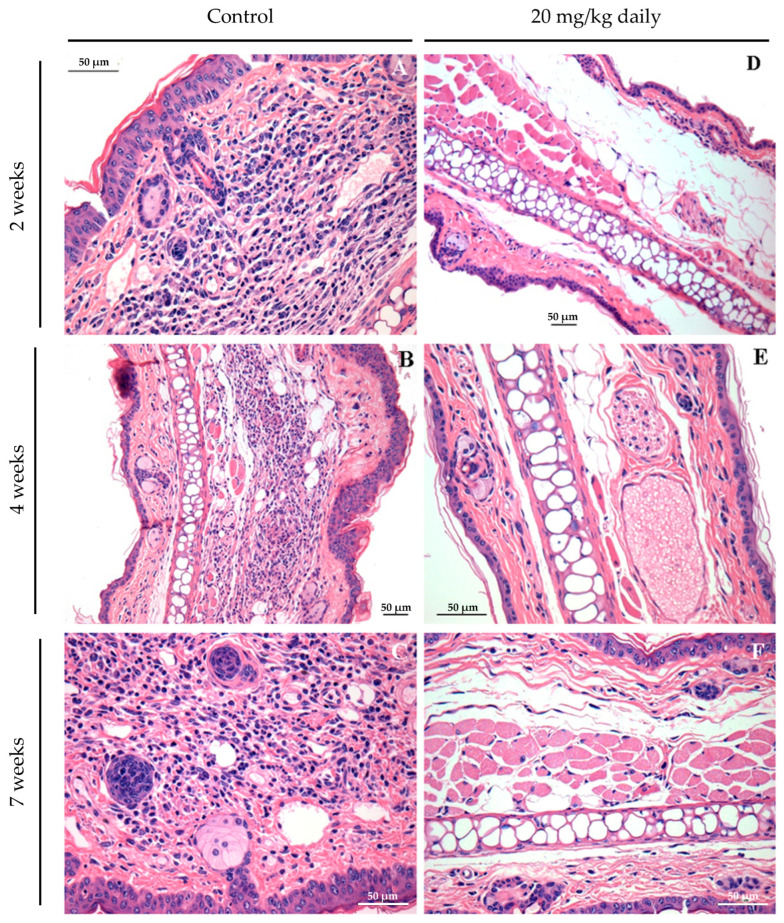
Long-term assessment of the inflammatory process in the ears of mice infected with *L. braziliensis* and treated intraperitoneally with 17-DMAG at later stages of treatment. Images represent inflammatory infiltrates in the lesion of BALB/c mice of the kinetic experiment described in Figure 7. After two (**A**,**D**), four (**B**,**E**), or seven weeks (**C**,**F**) after treatment. Control mice received 5% glucose solution intraperitoneally during the same time points. All animals were euthanized, and their ears were removed, embedded in paraffin, cut on a microtome, and stained with H&E for histopathological analysis. Histological sections were analyzed, and the images were obtained under observation in an optical microscope. Magnification 40×.

**Figure 9 pathogens-13-00630-f009:**
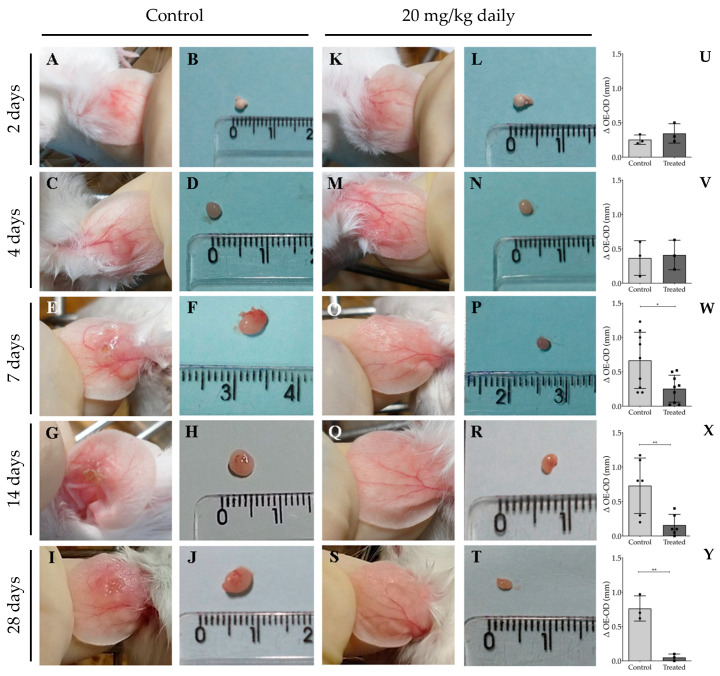
Macroscopic aspect of lesions in the ear and of lymph nodes from BALB/c mice infected with *L. braziliensis* intraperitoneally treated with 17-DMAG at early time points of treatment. BALB/c mice were infected in the ear with 5 × 10^5^ metacyclic form of *L. braziliensis* promastigotes. After two weeks of infection, animals were treated with 20 mg/kg/day of 17-DMAG for 2, 4, 7, 14, or 28 days (**K**–**T**). Control mice received 5% glucose solution intraperitoneally during the same time points (**A**–**J**). Images represent lesions (**A**,**C**,**E**,**G**,**I**) and lymph nodes (**B**,**D**,**F**,**H**,**J**) of BALB/c mice infected or infected and treated (lesions: (**K**,**M**,**O**,**Q**,**S**); lymph nodes: (**L**,**N**,**P**,**R**,**T**)). Animal lesions were measured from 2 to 7 days, then once a week, and ear sizes were expressed as the difference between the thickness of infected ears and contralateral ears after 2 (**U**), 4 (**V**), 7 (**W**), 14 (**X**), or 28 (**Y**) days of treatment. Unpaired *t* test, * *p* < 0.05; ** *p* < 0.01.

**Figure 10 pathogens-13-00630-f010:**
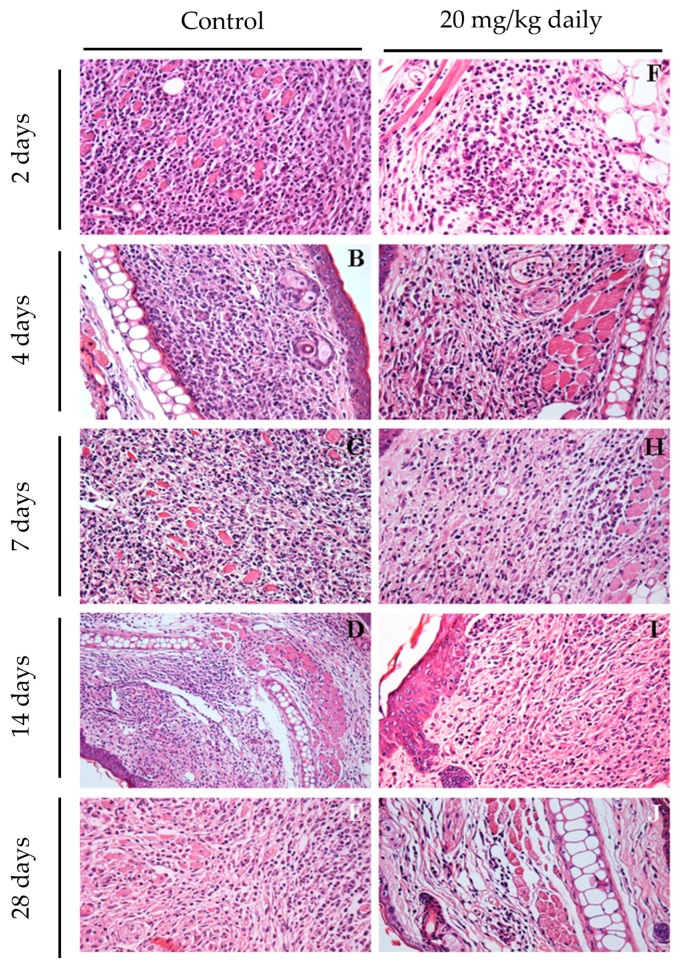
Early assessment of inflammatory process in ears of mice infected with *L. braziliensis* and treated with 17-DMAG intraperitoneally. Images represent inflammatory infiltrates in the lesion of BALB/c mice infected and treated as explained above. After 2 (**F**), 4 (**G**), 7 (**H**), 14 (**I**), and 28 days (**J**) of treatment. Control mice received 5% glucose solution intraperitoneally during the same time points (2 days: (**A**); 4 days: (**B**); 7 days: (**C**); 14 days: (**D**); 28 days: (**E**)). All the animals’ ears were removed, embedded in paraffin, cut on a microtome, and stained with H&E for histopathological analysis. The histological sections were analyzed, and the images were obtained under observation in an optical microscope. Magnification 400×.

**Figure 11 pathogens-13-00630-f011:**
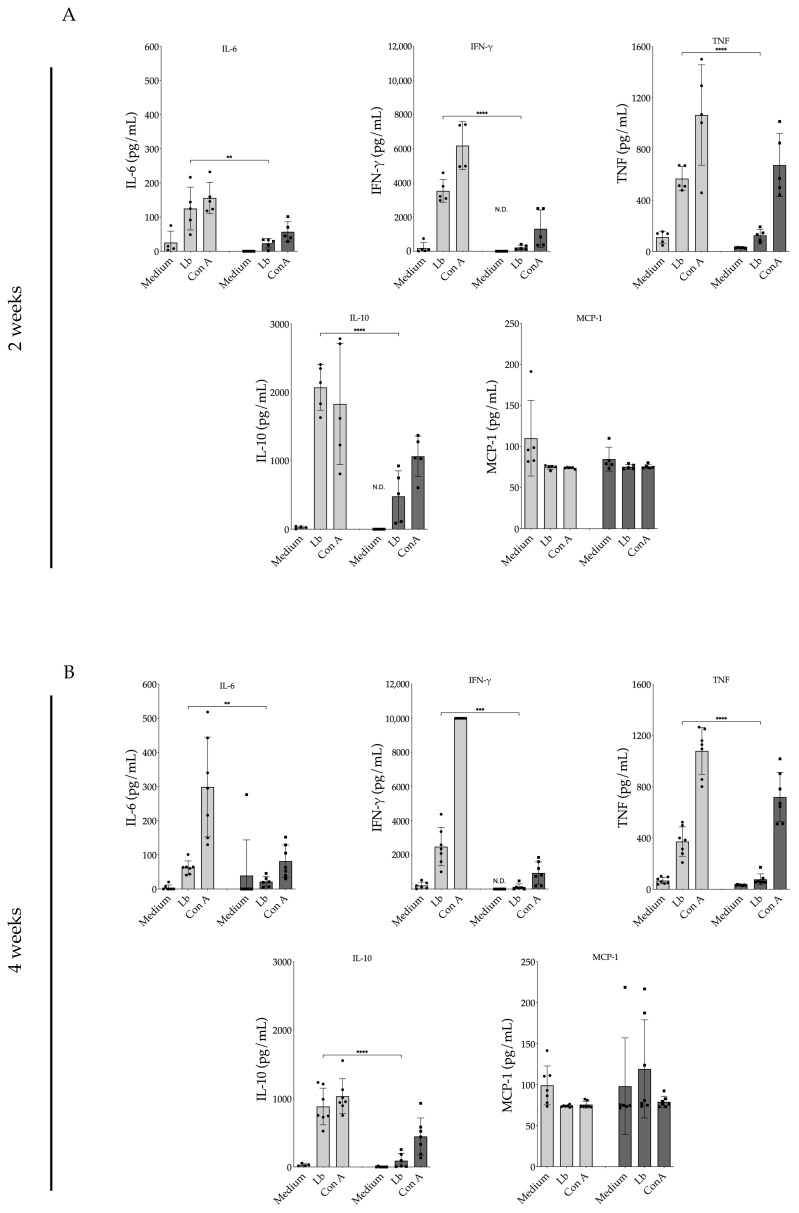
Production of pro-inflammatory cytokines by lymph node cells from *L. braziliensis*-infected BALB/c mice treated with 20 mg/kg/day of 17-DMAG. BALB/c mice were infected with 10^5^ promastigotes of *L. braziliensis* in the stationary phase and, after two weeks, were treated with 20 mg/kg/day with 17-DMAG. Control animals were treated daily with 5% glucose solution. The animals were euthanized after (**A**) two or (**B**) four weeks of treatment and the retroauricular lymph nodes were removed, homogenized, and cells were plated in complete RPMI medium at a concentration of 10^6^ cells per well. These cells were restimulated with *L. braziliensis* (5:1) or 5 µg/mL of (ConA) for 48 h. After this period, the supernatants were collected, and the production of the cytokines IL-6, IFN-γ, TNF, and IL-10, and of the chemokine MCP-1 in cells’ supernatant was quantified using a specific CBA kit for inflammatory cytokines according to the manufacturer’s instructions. Bars represent the mean value of the amount of cytokine released from control groups (light gray) and treated mice (dark gray) at 2- and 4-weeks post-treatment. Each black round symbol (●) represents the amount of cytokine produced by each control animal, and each square symbol (■) represents the amount of cytokine produced by each treated animal, in an experiment performed with seven mice per group. (Unpaired *t* test, ** *p* < 0.01; *** *p* < 0.001; **** *p* < 0.0001). N.D. = not detected.

## Data Availability

The original contributions presented in the study are included in the article, further inquiries can be directed to the corresponding author.

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
