# Peer review of "Intraperitoneal Administration of 17-DMAG as an Effective Treatment against Leishmania braziliensis Infection in BALB/c Mice: A Preclinical Study"

_pathogens, 2024, doi:10.3390/pathogens13080630_

Round 1

Reviewer 1 Report

Comments and Suggestions for Authors

Author Response

Dear Editor,

We want to express our gratitude for your and the referees' contributions. Your insights have been pivotal in refining our manuscript for potential publication in the Special Edition “Host Immune Responses in the Control of Leishmania Infection and New Forms of
Therapy for Tegumentary Leishmaniasis” in Pathogens.

We have thoughtfully incorporated all suggested changes and clarifications (highlighted in yellow in the new version of the manuscript) and into this rebuttal letter attached to this page. Referees’ constructive feedback has greatly aided us in enhancing the manuscript's quality and alignment with the journal's standards.

Thank you for your dedicated review.

Best regards,

Sincerely,

Patrícia Sampaio Tavares Veras, PhD

Corresponding author:
Patrícia Sampaio Tavares Veras
Address: Laboratory of Parasite–Host Interaction and Epidemiology, Gonçalo Moniz Institute, FIOCRUZ. Rua Waldemar Falcão, 121, Candeal - Salvador/BA, Brazil. Zip Code: 40296-710
Phone: +55 71 98122-0451
Email: patricia.veras [email protected]; [email protected]

Reviewer 2 Report

Comments and Suggestions for Authors

Article: Intraperitoneal Administration of 17-DMAG as an Effective Treatment Against Leishmania braziliensis Infection in BALB/c Mice: A Preclinical Study

Journal: Pathogens

Manuscript ID: Pathogens-3105614

By Kercia P. Cruz et al.

Comments to the authors

Leishmaniasis is a neglected disease that continues to pose a public health problem. This study focused on 17-DMAG, an Hsp90 inhibitor, and highlights its efficacy against L. braziliensis infections both in vitro and in vivo. The results showed a potential effect of 17-DMAG as a therapeutic agent for treating cutaneous leishmaniasis.

I consider that the general aim of the manuscript is very interesting and the methods used appear robust, including both in vitro and in vivo approaches to assess treatment effectiveness. Overall, this work seems well structured and provides valuable information on the potential of 17-DMAG as a treatment for cutaneous leishmaniasis. Nevertheless, I consider that some points must be revised and some questions still needing the answer. Hereafter the authors will find my comments.

    Minor points that need to be corrected

Throughout the article, the words in vivo and in vitro are not italicized and must be corrected in italics.

    Major points and questions

1-     In Material and Methods §:

-       (Line 135-137): There are limitations inherent to this method of enriching the metacyclic forms of L. braziliensis with lectin. These limitations may include variations in enrichment efficiency that may affect the quality of in vivo infection.

-       (Line 197): Although the authors mentioned the use of triplicates, it would be useful to provide more details on the repetition of the experiment to ensure the reproducibility.

-       (Lines 217-218): The authors opted for a ratio of 10:1 and 24 hours of contact between the parasite and the macrophages. It’s known that 4 hours of contact would have been sufficient. Don't you think that 24 hours with this ratio would negatively impact cell viability and therefore your results?

-       In 2.7 section (line 233), there are some points to improve:

I suggest long-term observation. Indeed, longer-term post-treatment observation could provide additional information on the durability of therapeutic effects.

Do you think that intraperitoneal administration is the most appropriate for treating a dermal infection?

-       In 2.9 section (line 267): To analyze cytokine production, it is recommended to assay cytokines at multiple time intervals, including early times and optionally 48 hours, to obtain a complete picture of cytokine production and release in culture supernatants.

-        

2-     Below are some general questions:

This work showed the effect of 17-DMAG against L. braziliensis. Responses to treatments may vary depending on parasite strains.

Leishmaniasis is a complex disease with different parasite species and variations in host immune response. It is important to consider this complexity when evaluating the effectiveness of a potential treatment.

Author Response

Dear Editor,

We want to express our gratitude for your and the referees' contributions. Your insights have been pivotal in refining our manuscript for potential publication in the Special Edition “Host Immune Responses in the Control of Leishmania Infection and New Forms of
Therapy for Tegumentary Leishmaniasis” in Pathogens.

We have thoughtfully incorporated all suggested changes and clarifications (highlighted in yellow in the new version of the manuscript) and into this rebuttal letter attached at this page. Referees’ constructive feedback has greatly aided us in enhancing the manuscript's quality and alignment with the journal's standards.

Thank you for your dedicated review.

Best regards,

Sincerely,

Patrícia Sampaio Tavares Veras, PhD

Corresponding author:
Patrícia Sampaio Tavares Veras
Address: Laboratory of Parasite–Host Interaction and Epidemiology, Gonçalo Moniz Institute, FIOCRUZ. Rua Waldemar Falcão, 121, Candeal - Salvador/BA, Brazil. Zip Code: 40296-710
Phone: +55 71 98122-0451
Email: patricia.veras [email protected]; [email protected]
